# Structural variability of multifunctional proteins indicates frequent stochastic evolution of protein oligomers
György Abrusán [1] ✉ & Aleksej Zelezniak[1,2,3]

Recently, it has been suggested that the evolution of many protein homomer complexes follows a neutral pattern, with little effect on their biochemical function. One of the strongest arguments in support of this hypothesis is the observation that homologous enzymes with the same catalytic function can have different quaternary structures in various species. However, in the case of proteins with multiple functions ("moonlighting" proteins), this pattern can also have an adaptive explanation if quaternary structure is responsible for their variable, non-canonical functions. To test whether moonlighting can be responsible for the variability of quaternary structure, here we examine the opposite of the "same function–multiple structures" pattern, and test whether orthogroups of moonlighting (multifunctional) and non-moonlighting proteins have similar quaternary structure variability. We show that there is very little association between moonlighting and homomer quaternary structure diversity, which is in agreement with the neutral expectation and the hypothesis that many homomers might be adaptive by shaping the biophysical characteristics of the cell and cytoplasm, rather than the biochemical function of the protein.

Most proteins function as part of a protein complex, which can be either a heteromer, assembled from several different proteins, or a homomer, assembled from multiple instances of the same protein. Even though homomers are very common[1], especially in prokaryotes, the functional relevance of homomer formation is unclear. Currently, there are two views on the factors that govern the evolution of homomers. The traditional view is that such complexes are adaptations enabling the biochemical, or some other function of the protein[2], while more recently it has been suggested that their evolution follows a neutral pattern[3] and frequently might be adaptive only at the level of maintaining cellular homeostasis[4], rather than at the level of protein function.

Several arguments support both views, and currently, it is not known what fraction of homomers evolves in a neutral/stochastic or adaptive manner. The adaptationist view is supported by the facts that many homomers can function only in their oligomeric form; interfaces are frequently conserved[5], quaternary structure similarity depends on sequence similarity[6]; mutations in protein-protein interfaces are often pathogenic[7]; complex formation frequently has a role in the dynamics, allostery, and the regulation of protein function[2,8–10] (for example only one of the topologies/conformers might be active; among allosteric proteins dihedral symmetry is enriched); and can also result in higher stability of proteins[11].

However, some of these characteristics can also be the result of constructive neutral evolution (CNE)[12–15], which can result in an "entrenchment" of interfaces and the evolution of stable, obligate complexes[12]. The neutral view is also supported by several arguments: homologous enzymes with the same function frequently have different quaternary structures[3,16–18]; in a fraction of homomers (which in some dimers might be as high as 35%[19]) their monomeric subunits can perform the same function as the full complex[12,20]; novel quaternary structures can emerge due to a small number of point mutations[21]; in homomers with binding sites restricted to a single subunit the evolution of ligand binding/function follows the same pattern as in monomers[22]; the hydrophobicity of homomer interfaces is only weakly affected by the strength of purifying selection[19]; and that the turnover of subunits is biased towards subunit gains[4], as predicted by constructive neutral evolution and the entrenchment of interfaces[12].

The neutral and adaptationist views are not mutually exclusive, and currently, it is unclear to what degree stochastic processes shape the complexome. The same complexes might be affected by both adaptive and stochastic factors; for example, if the interface of a dimer is adaptive, while the higher-order multimers of such dimers (e.g., tetra-, hexa-, octamers) evolve due to a neutral/stochastic process. Additionally, even though a substantial QS variability that is not linked to protein function clearly exists,

[1]Randall Centre for Cell and Molecular Biophysics, School of Basic and Medical Biosciences, King's College London, New Hunt's House, Great Maze Pond, London, UK. [2]Department of Life Sciences, Chalmers University of Technology, Gothenburg, Sweden. [3]Institute of Biotechnology, Life Sciences Centre, Vilnius University, Vilnius, Lithuania. ✉e-mail: gyorgy.abrusan@kcl.ac.uk

**Fig. 1 | Outline of the hypothesis tested. A** In many species, proteins with the same function (e.g., enzymes) have different quaternary structures (QS). This was interpreted as a sign of neutral evolution of QS, however, in the case of moonlighting proteins, differences in QS may be the result of their additional, non-canonical functions, which can vary depending on species. **B** Neutral and adaptive explanations of QS evolution result in different predictions for its association with moonlighting: if moonlighting does not result in higher QS variability within homologous proteins (Case 1), that supports the hypothesis that in homomers the evolution of QS is strongly influenced by stochasticity, or selection at a different level than protein function, like the maintenance of cellular homeostasis. In contrast, if QS evolution of homomers is driven by the evolution of protein function, i.e., it is adaptive, higher QS variability would be expected in moonlighting proteins (Case 2).

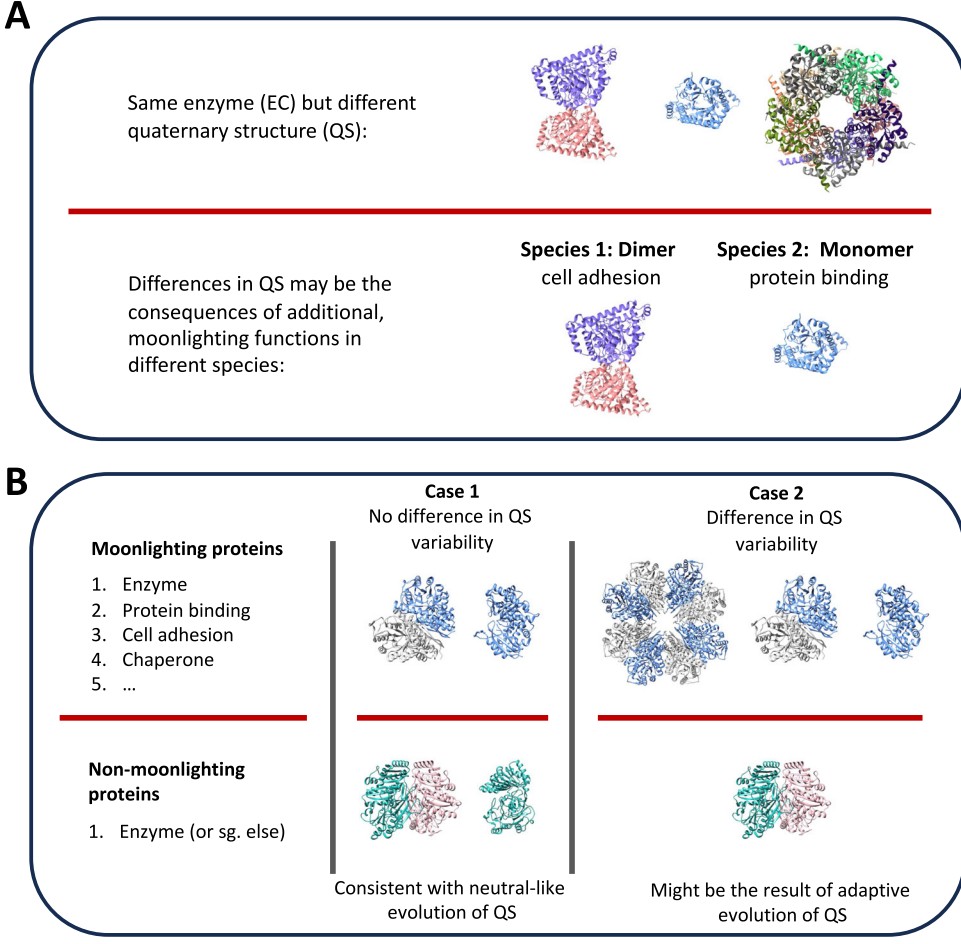

in many species, especially autotrophs, hydrophobic residues can have a higher synthesis cost than other residues[23], and in such species, the existence of hydrophobic interfaces is probably not strictly neutral[4], but might be adaptive in maintaining cellular homeostasis[4]. Thus, it is probably more appropriate to use the neutral-like, or stochastic evolution terms for such quaternary structure variability.

In this paper, using multifunctional, so-called "moonlighting" proteins[24], we test to what degree the quaternary structure of homomers is shaped by protein function and stochasticity (Fig. 1). The observation that enzymes with the same functions frequently have different quaternary structures is one of the strongest arguments for the role of stochasticity in the evolution of quaternary structure[3] (Fig. 1A). However, several ancient enzymes are also known to be moonlighting proteins[25,26], and can have diverse additional functions, like chaperone activity, scaffolding, cell adhesion or others[25,27]. Proteins can "moonlight" via several different mechanisms, ranging from changes in subcellular localisation, changes in binding partners, changes in the topology of their fold (including disordered regions), and changes in oligomerisation are also known[25,27]. Examples include peroxidases that are enzymes as dimers, chaperones as decamers[27]; fructose 1,6-biphosphate aldolases, which show large structural and functional variability across different species[28], hexokinases[29], or morpheeins, for example, porphobilinogen synthase[27].

In such enzymes, the variability of quaternary structure in their phylogenies[3,4,17] might be the result of the variability of their additional, moonlighting functions (Fig. 1A). However, the neutral and adaptive theories make different predictions on the evolution of quaternary structure for moonlighting and non-moonlighting protein families (orthogroups): the neutral view predicts that function does not influence quaternary structure, thus its variability will not be different between protein families that moonlight and the families that do not (Case 1, Fig. 1B). In contrast, the adaptive explanations predict that quaternary structure variability will be higher in the moonlighting protein families (Case 2, Fig. 1B).

Here, by analysing ~20k proteins from the Protein Data Bank (PDB), we examined which of these two predictions is supported by the available structural data. We found that there is no or just a weak association between moonlighting and homomer quaternary structure diversity, indicating that the evolution of quaternary structure in homomers is frequently stochastic (i.e., supporting Case 1). These results also suggest that, in general, moonlighting proteins not so much drive but rather utilise the preexisting variability of quaternary structure, at least when QS variability between different species and proteins is considered.

## Results

### Characteristics of the proteins used

Using the Protein Data Bank, we compiled a set of proteins with a high-quality structure (see Methods), in total 21,341. These proteins were classified into 4181 orthogroups, using the eggNOG-mapper tool[30] and their Pfam domain annotations (see Methods), which reduced their number to 19,783, of which 12,798 are homomers and 6985 are monomers (Supplementary Data 1). The number of known, experimentally validated moonlighting proteins is low because they are usually detected only accidentally, and their estimated numbers in the human proteome vary significantly, from 3%[31] to more than 30%[32]. We compiled a set of 759 moonlighting proteins from the MoonProt[33] and the MoonDB[34] databases (see Methods). However, this small number of proteins is only the tip of the iceberg, and the examination of particularly well-studied protein families like enolases with many known moonlighting proteins suggests that within the orthogroups that do contain some moonlighting proteins, they are widespread and relatively evenly distributed across the phylogeny (see Fig. 2A, B). Therefore, instead of directly comparing known moonlighting proteins with proteins

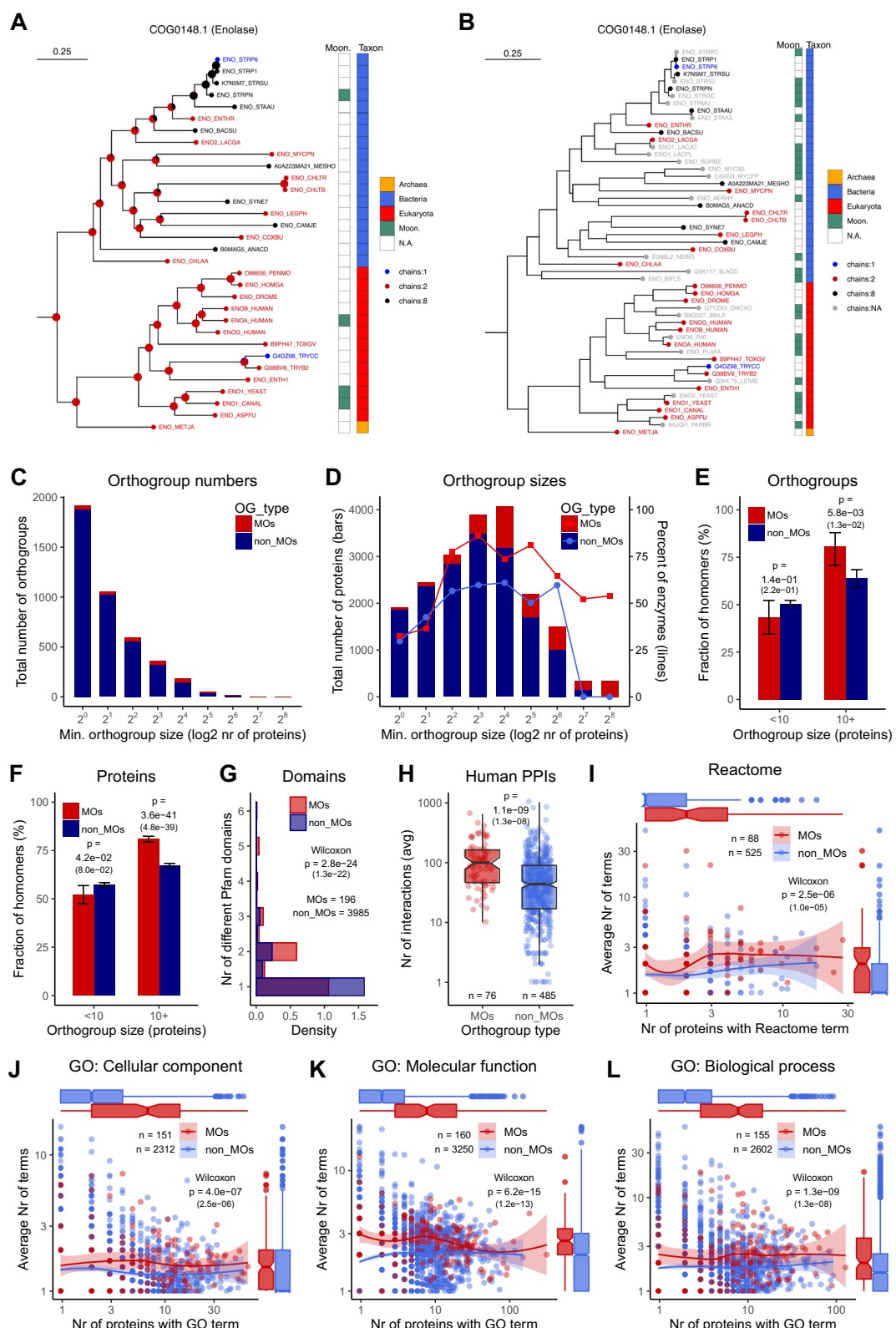

that are not known to be moonlighting, we used known moonlighting proteins as "markers" to identify protein families characterised by multifunctional proteins. We identified the list of orthogroups that have at least one known moonlighting protein, and compared them with the orthogroups that have no known moonlighting proteins. The moonlighting orthogroups (MOs) were identified by adding the 759 moonlighting proteins to the set of proteins with known structures, and re-running the orthogroup identification procedure (eggNOG-mapper). This has resulted in 196 orthogroups that have high-quality homomers or monomers in the PDB (MOs), and at least one moonlighting protein (which may not necessarily have a structure in the PDB), and 3985 orthogroups without moonlighting proteins (non-MOs). MoonDB includes also predicted moonlighting proteins, where GO terms are utilised in the prediction. Thus, to ensure independence, in all analyses that compare GO terms of MOs and

**Fig. 2 | Orthogroups (OGs) having moonlighting proteins are consistently annotated with more functions. A** Phylogeny of the enolases, using proteins with a high-coverage PDB structure. Pie charts at the nodes indicate the probability of subunit number; green squares indicate the known moonlighting proteins in the tree; the taxonomic domain of the proteins is indicated with blue, red, and orange squares. **B** The same phylogeny, with all known moonlighting enolases added; the ones not having high-coverage structures are indicated with grey. Moonlighting proteins are scattered through the entire tree, suggesting that many of the proteins not annotated as moonlighting on panel A also have multiple functions in this orthogroup. **C** Distribution of orthogroup sizes. The majority of orthogroups are small, with less than four proteins; nevertheless, most proteins are in the orthogroups with 8 or more proteins (see **D**). **D** The total number of proteins in orthogroups of different size. As reported previously, moonlighting orthogroups

(MOs) are characterised by a higher frequency of enzymes than non-MOs. **E** The fraction of orthogroups with a homomer majority, depending on orthogroup size. (Tests of proportions, whiskers show 95% confidence intervals (CI)). **F** The fraction of homomers in all proteins, depending on orthogroup size. (Tests of proportions, whiskers show 95% CI). **G** The average number of different Pfam-domains per sequence is significantly higher in the MOs than in non-MOs (Wilcoxon rank sum test). **H** The average number of human PPIs in an orthogroup is significantly higher in the MOs than in the non-MOs. **I** The average number of Reactome terms per protein is significantly higher in the MOs than in the non-MOs. **J–L** Similarly to Pfam domains, PPIs, and Reactome, the average number of GO terms per protein is significantly higher in the MOs than in the non-MOs, in all three GO categories (See also Supplementary Figs. 1 and 2). On all panels, numbers in parentheses are $p$ values corrected for multiple testing.

non-MOs, we used a reduced set of proteins for MO identification that did not include predicted proteins from MoonDB. This resulted in 161 MOs, while the number of non-MOs was not changed.

In total, 433 moonlighting proteins could be categorised into the 196 moonlighting orthogroups (Supplementary Data 2), of which 229 have any structure in the PDB, and only 130 have a quaternary structure with high-coverage of the protein: 66 homomers, 10 monomers, and also 54 that are part of a heteromer. The size of the orthogroups is highly variable; the majority of orthogroups are small, with less than four proteins (Fig. 2C); however, most proteins, especially in MOs, are present in the larger orthogroups. In total, 3113 proteins are present in MOs, of which 2808 are present in the orthogroups with 8 or more proteins, and 16670 and 9572 in non-MOs, respectively (Fig. 2D). Enzymes have higher frequencies in MOs than in non-MOs (Fig. 2D). In orthogroups with 10 or more proteins, i.e., the ones that were used in the phylogenetic analyses (see below), homomers have significantly, although not dramatically, higher frequencies than monomers (80% vs. 63–67%, Fig. 2E, F).

## Proteins in moonlighting orthogroups are characterised by more functions

As the vast majority of proteins in MOs are not present among the proteins from the moonlighting databases, we tested whether proteins in the MOs are generally characterised/annotated by more functions than proteins in non-MOs. We examined five protein characteristics: the number of different Pfam domains per protein, the number of human protein-protein interactions (PPIs), Reactome annotations, Gene Ontology annotations, and the number of subcellular locations from the Human Protein Atlas[35]. We found that proteins in MOs have significantly more different Pfam domains than proteins in non-MOs (Fig. 2G). Additionally, within the orthogroups that contain human proteins, proteins in MOs have a significantly higher number of PPIs than those in non-MOs (Fig. 2H). Similarly, both the Reactome (Fig. 2I) and Gene Ontology annotations (Fig. 2J–L) indicate a consistent difference between the two groups, indicating that proteins in MOs are generally annotated with more functions than proteins in non-MOs. However, within MOs, there is a significant difference between known moonlighting proteins and proteins for which information on moonlighting is not available (Supplementary Fig. 1); thus, we repeated the analysis with the known moonlighting proteins excluded from MOs. The results indicate that the patterns are not substantially influenced by the small number of known moonlighting proteins in MOs (Supplementary Fig. 2). We also repeated the GO analysis using subsets of GO terms (Supplementary Fig. 3, see also "Methods"): with the terms with IEA evidence codes excluded (Supplementary Fig. 3A–C), and using only terms with experimental evidence (Supplementary Fig. 3D-F), which show a smaller, nevertheless significant difference. Surprisingly, even though moonlighting is frequently thought to result in changes of subcellular locations[25], and this is supported by the GO Cellular Component terms (Fig. 2J), the analysis of Human Protein Atlas (HPA) subcellular locations shows no clear differences between moonlighting and non-moonlighting proteins (Supplementary Fig. 4). Besides simply incomplete knowledge or errors in the HPA (e.g., due to off-target antibody binding), a possible explanation of this pattern is that

these locations are likely to include also the cases of protein mislocalization[36,37] present in the HPA cell lines.

Taken together, these results indicate that, on average, proteins in MOs have more functions than proteins in non-MOs, even when only proteins that currently lack experimental evidence of moonlighting are considered (Supplementary Fig. 2). Thus, the small number of known moonlighting proteins can be used as markers of protein families characterised by multifunctionality (instead of using them directly), which allows larger-scale comparisons of quaternary structure, based on thousands of proteins, rather than dozens.

## The rate of quaternary structure change is similar in moonlighting and non-moonlighting phylogenies

Next, we quantified the rate of quaternary structure change in the phylogenies of the orthogroups, using orthogroups with 10 or more proteins. We made rooted phylogenetic trees for each orthogroup, and estimated the quaternary structure at their nodes (including the root) with a discrete maximum likelihood method, using the known quaternary structure of the proteins at the leaves (see Figs. 2A and 3A, and "Methods"). In addition, in a separate analysis, the ancestral states of the nodes were also estimated with maximum parsimony (Supplementary Fig. 5). The rate of QS change was determined as the number of parent-child nodes (including the leaves of the tree) that do not have the same QS, divided by the total number of the nodes of the tree. As the rate of QS change in an orthogroup is significantly affected by the type of the root of the tree (i.e., homomer or monomer)[4], and MOs have higher homomer frequencies (Fig. 2E, F), similarly to[4], we analysed trees with homomer root and monomer root separately.

In the case of trees with homomer root, we found no significant difference, neither for the overall rate of QS change (Fig. 3C), nor for subunit gains or losses (Fig. 3D). The average interface area of proteins in MOs is higher than in non-MOs (Fig. 3E), but there is no difference in the average distance of proteins from the root (Fig. 3F). As interface size is likely to influence the rate of QS changes and interface evolution (i.e., the magnitude of the overlap in interfaces between homologues), we also compared MOs and non-MOs using interface size as a covariate; we found no significant difference between the two for the rate of QS change (Fig. 3G), and also no difference in the average interface-overlap within the orthogroups (Fig. 3H). Ancestral character estimates based on maximum parsimony (Supplementary Fig. 5) show a similar pattern to maximum likelihood, i.e., a lack of difference between MOs and non-MOs. However, among homomers with small, less than 1000 Å² interfaces, crystallographic artefacts are frequent[38]. To test whether possible crystallographic artefacts influence the results, we repeated the analysis with such homomers being included among the monomers (Supplementary Fig. 6). The results show no significant differences in the rate of QS evolution, even when interface area is added as a covariate (Supplementary Fig. 6E).

Similarly, we found no differences in the rate of QS evolution in the trees with monomer root (Fig. 3I–L and Supplementary Fig. 6G–J), however, the number of such trees in MOs is very small ($n = 14$). Taken together, these results suggest that there are no clear differences in the rate of QS

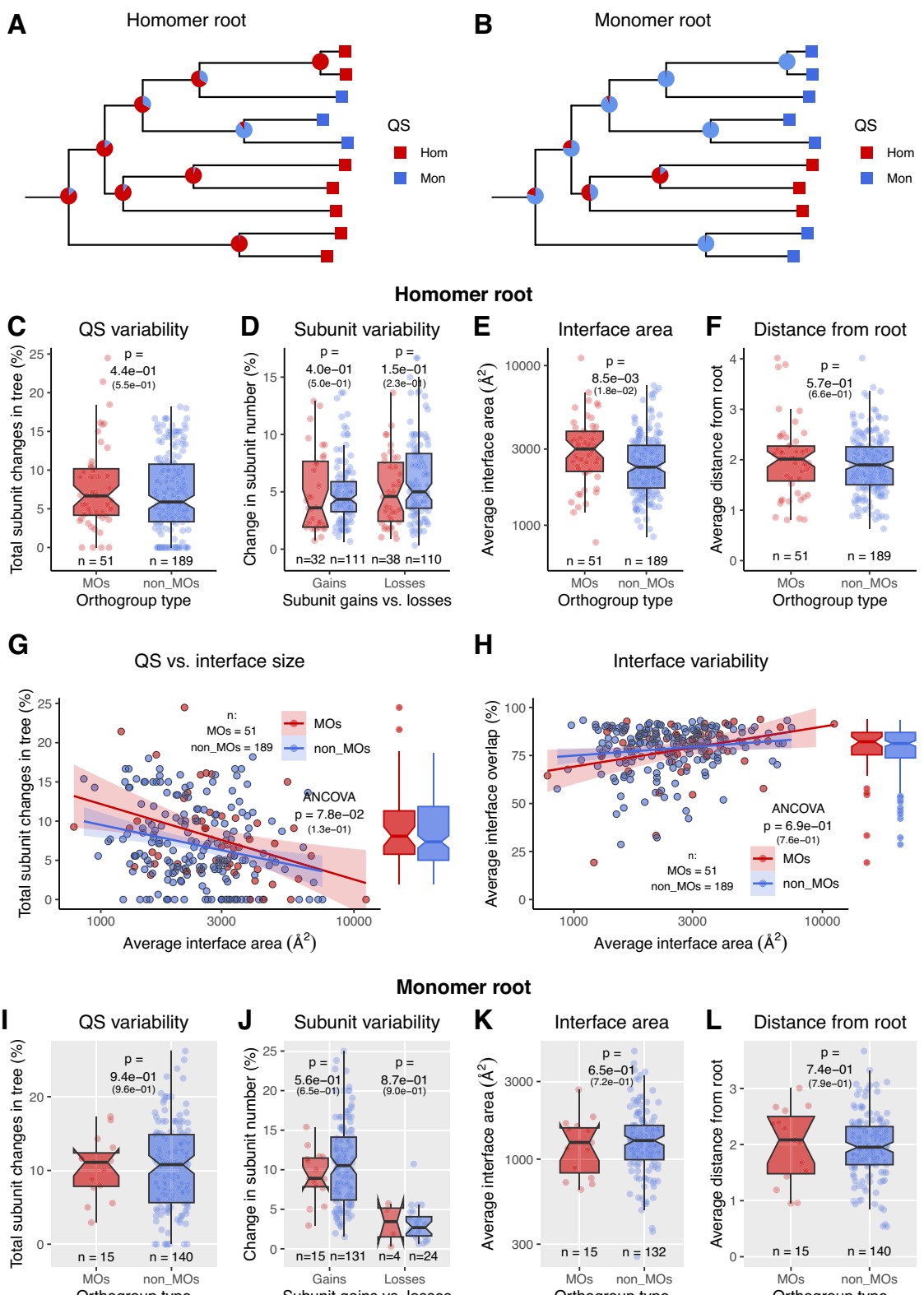

**Fig. 3 | The rate of change in quaternary structure is similar in phylogenies of MOs and non-MOs. A**, **B** Examples of trees with a homomer and monomer root (note that the topology of the trees is similar). Only trees with 10 or more proteins were used in the analyses. **C** In the trees with homomer root, the total frequency of QS changes is not significantly different in MOs and non-MOs. **D** Similarly, the frequency of subunit gains and losses is also not different in the trees of MOs and non-MOs. **E** The average interface area of homomers in MOs is significantly higher than in non-MOs. **F** The average evolutionary distance from the root is similar in MOs and non-MOs. **G** The rate of QS change remains similar when the average interface area of the orthogroups is added as a covariate. **H** The average interface overlap in the two groups is also similar, suggesting that new interfaces are evolved at a similar rate in MOs and non-MOs. **I–L** In the case of trees with monomer root, no significant differences could be detected between MOs and non-MOs, however, the power of the analyses is very low, due to the few such trees in MOs. On all panels, numbers in parentheses are *p* values corrected for multiple testing. Except for **G** and **H**, *p* values were obtained with Wilcoxon rank sum test.

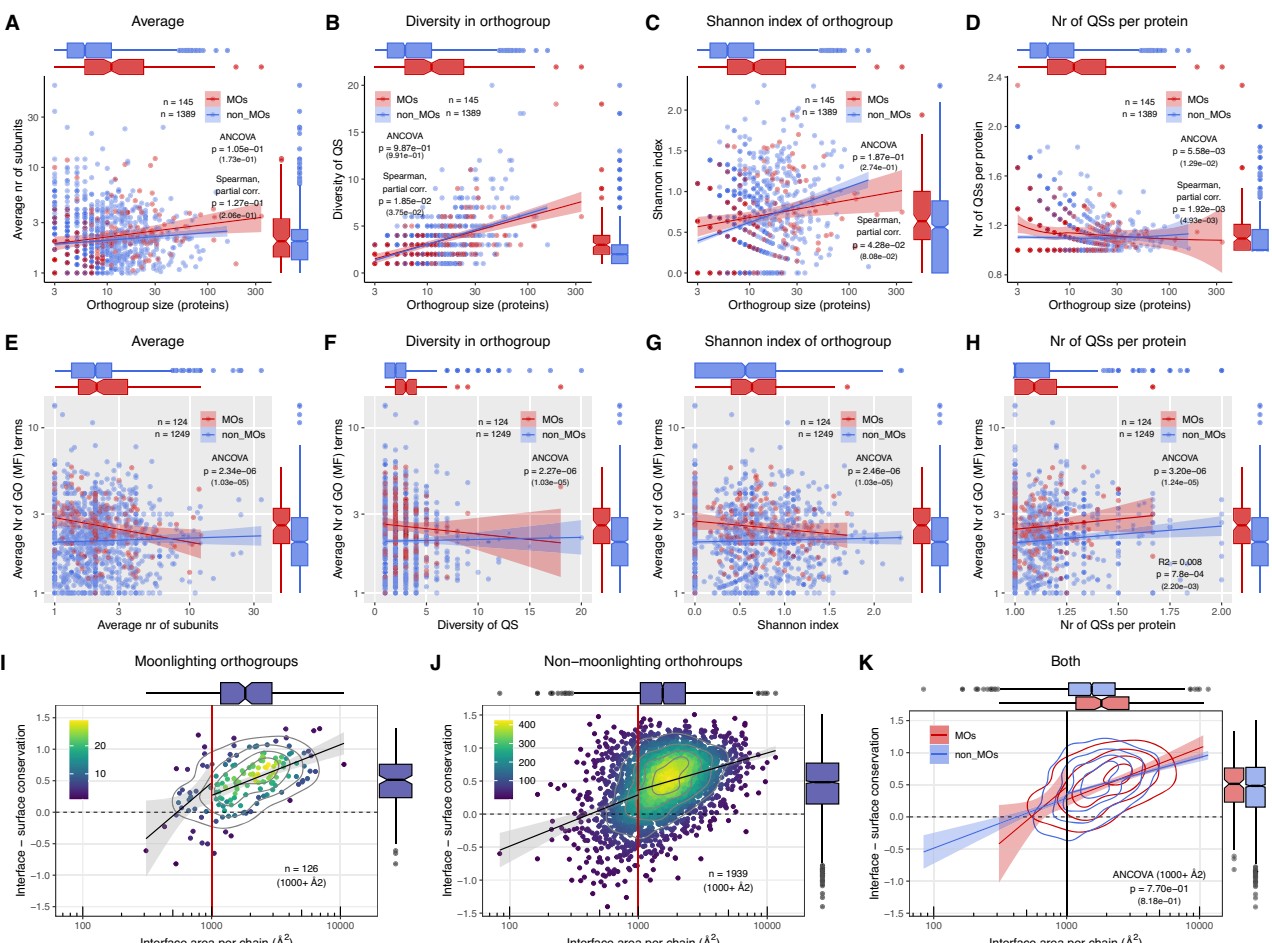

**Fig. 4 | Quaternary structure diversity metrics do not explain the functional difference between MOs and non-MOs. A–D** The average number of subunits, their diversity, and the Shannon index do not differ significantly between MOs and non-MOs. Only the number of different QS per protein shows a small, nevertheless significant difference between the two groups (**D**); however, it disappears or weakens when homomers with small interfaces (<1000 Å²) are included among the monomers (Supplementary Fig. 7). **E–H** The correlations between QS diversity metrics and the number of Molecular Function GO terms also indicate that QS diversity does not explain the difference in functional diversity between the two groups (see Supplementary Fig. 9 for Cellular Component and Biological Process GO terms). In addition, we find no clear positive correlation between diversity metrics and the average number of GO terms, suggesting that on evolutionary timescales, changes in QS and the evolution of function are frequently independent. The exception is the number of different QS per proteins (**H**, see also Supplementary Fig. 8D, H), where we could consistently detect a weak, nevertheless significant positive correlation with the number of GO terms, which explains 1–2% of the variance in the GO terms. **I–K** Conservation of interfaces in the MOs and non-MOs. Each dot represents the average interface conservation (compared to the surface) and average interface size of homomers in an orthogroup. No significant difference could be detected between the MOs and non-MOs (**K**, see also Supplementary Fig. 12). On all panels, numbers in parentheses are *p* values corrected for multiple testing.

evolution between MOs and non-MOs, although the number of trees with MOs is relatively low, and a weak trend might be detected with more statistical power (Fig. 3G).

## Quaternary structure diversity does not explain the functional differences between MOs and non-MOs

The phylogenetic analyses above could not be performed for small orthogroups, and the majority of orthogroups have less than 10 proteins (Fig. 2C). Thus, to include more orthogroups in the analysis, we also performed comparisons of MOs and non-MOs using simpler metrics of QS variability, which we applied to orthogroups with a minimum of three proteins. We used four metrics: (1) the average number of complex subunits in the orthogroup, including monomers; (2) QS diversity, measured as the number of different quaternary structures that are present in an orthogroup; (3) The Shannon index, which is based on the frequency of quaternary structures in the orthogroup (see "Methods"), (4) The average number of different QSs per protein in an orthogroup, which was calculated using all entries with different QS that are available in the PDB for a particular protein (see "Methods").

These indices are affected differently by the characteristics of the data, and any eventual errors in QS assignment in the PDB[39], which inevitably affect the dataset: QS diversity (2) can be substantially influenced by one or two proteins in the orthogroup, because it does not take into account their frequency, while the Shannon index (3), which uses also the frequency of any given QS is much less affected by this, at least in the larger orthogroups. The average number of QS per protein (4) is likely to be overestimated for many proteins, which normally function as homomers, because obligate homomers sometimes also have monomeric entries deposited in the PDB, while for proteins with variable quaternary structures, existing PDB entries may not cover their entire structure space.

The comparison of MOs and non-MOs indicates that there is no, or just very little, difference between the two groups (Fig. 4). The average number of subunits, QS diversity, and the Shannon index are not significantly, or just marginally different (Fig. 4A–C) when the size of the orthogroup is used as a covariate, both with a parametric (ANCOVA) and with a non-parametric (Spearman partial correlation) method. The average number of QS per protein shows a significant, nevertheless small difference (Fig. 4D), however when homomers with small, less than 1000 Å² interfaces

are included among the monomers, the difference largely disappears (Supplementary Fig. 7D). Since some of the regressions show deviations from regression assumptions (homoscedasticity, normally distributed residuals), we also compared the two groups with the axes swapped (Supplementary Fig. 8), which consistently show that at the same QS diversity level, MOs have more proteins.

We also examined the dependence of GO-term richness on the four diversity metrics (Fig. 4E–H and Supplementary Fig. 9). We found that the consistent, highly significant difference between MOs and non-MOs is not explained by any of the QS diversity metrics, in any of the three GO categories (Molecular Function: Fig. 4E–H, Cellular Component and Biological Process: Supplementary Fig. 9). The inclusion of homomers with small interfaces (<1000 Å$^2$) among monomers also does not influence this pattern (Supplementary Fig. 7E–H). Surprisingly, in the case of average subunit number, QS diversity, and Shannon index, there is no positive correlation between QS diversity and GO-term richness. However, we found significant positive correlations in the case of the average number of QSs per protein (metric 4), which is weak, and explains only 1–2% of variance depending on the term used, nevertheless, it is consistent across the three different GO categories (Fig. 4H and Supplementary Figs. 9D and S8H). Since the average number of QSs per protein is likely to be affected by erroneous QS entries in the PDB, particularly monomer entries of proteins that in reality are multimers, it is very likely that the weak positive correlation we observe is an underestimate.

The diversity of QSs is also likely to be influenced by the divergence of proteins within orthogroups; in orthogroups with closely related proteins, QS is likely to be more conserved, while in orthogroups containing very distantly related proteins, QS is likely to be more variable, and moonlighting proteins are enriched among ancient proteins with conserved (frequently metabolic) functions[25]. Using the orthogroups with phylogenetic trees, we examined whether a difference in diversity emerges when the evolutionary distance within an orthogroup is used as a covariate (Supplementary Fig. 10). The results show that, as expected, MOs are characterised by a somewhat lower average distance within an orthogroup than non-MOs, and QS diversity scales positively with the average distance of proteins within an orthogroup (Supplementary Figs. 10B and S10C). However, except for the number of subunits (Supplementary Fig. 10A, see also Fig. 2E, H), the diversity–evolutionary distance relationship is not different in MOs and non-MOs (Supplementary Fig. 10B–D), while a clear difference in GO-Molecular Function terms could be detected, which is not explained by the diversity metrics (Supplementary Fig. 10E–H).

In addition to the comparisons between different orthogroups, we also compared the moonlighting and non-moonlighting proteins within MOs (Supplementary Fig. 11). We used the average number of subunits and the number of QSs per protein metrics only, because these are meaningful also when they are calculated only for a single protein. Unlike for GO terms, where a clear difference exists between moonlighting and non-moonlighting proteins (Supplementary Fig. 1), we found no difference in these QS metrics.

Taken together, these data are consistent with the results of the phylogenetic analysis and suggest that the observed functional differences between MOs and non-MOs are only minimally explained by quaternary structure variability of the orthogroups.

### The conservation of interfaces is not different in MOs and non-MOs

Finally, we examined whether the conservation of interfaces (compared to the conservation of solvent-accessible surface) differs between the proteins of MOs and non-MOs. We used the summary files from the ConSurf database[40] to calculate the difference in conservation between the interface and surface of every protein. The two groups were compared in two ways, by using the average interface area and average conservation of the orthogroups (Fig. 4I–K and Supplementary Fig. 10I–K), and by clustering the proteins at 30% sequence similarity, and using the cluster centroids in

the comparison (Supplementary Fig. 12). This latter procedure can result in multiple clusters within a single orthogroup. Both methods show that interface conservation is not significantly different between MOs and non-MOs (Figs. 4K and S9K and S11C).

## Discussion

Probably the most intuitive argument for the neutral-like evolution of homomers is the observation that homologous enzymes with the same function can have very different quaternary structures[3]. However, protein moonlighting can provide an alternative, adaptive explanation for this pattern, if quaternary structure is responsible for the non-canonical (e.g., non-enzymatic) functions of the protein (Fig. 1A), and many ancient, core metabolic enzymes are known to moonlight[29].

Here, we performed the opposite of the "same function—multiple QS" test, and examined whether the "multiple functions—same QS" pattern holds (Fig. 1B), which would support the hypothesis that a substantial fraction of QS variability is neutral from the perspective of protein function. Our findings are in agreement with the predictions of a stochastic QS evolution: while we could detect clear differences in the functional annotation of moonlighting and non-moonlighting orthogroups (Fig. 2), we found little or no differences in the variability/diversity of their quaternary structures (Figs. 3 and 4). The QS diversity metric that is most consistently correlated with functional diversity is the number of different quaternary structures per protein (Fig. 4 and Supplementary Fig. 9); however, even this metric explains only a small fraction (1–2%) of GO-term variance, and does not explain the difference between MOs and non-MOs. The strength of this association is likely to be significantly underestimated, though, due to QS errors in the PDB, but also due to the incomplete sampling of the structure space, i.e., that not all functional forms of homomers might be present in the PDB. In addition, in orthogroups with 10 or more proteins, the number of subunits is higher in MOs than in non-MOs (Fig. 2E–H and Supplementary Fig. 10A), suggesting that in some cases simply the larger size of a homo-oligomer, rather than its structural variability, might contribute to moonlighting functions.

While the number of known, experimentally characterised moonlighting proteins is low, the availability of genome-wide and high-throughput datasets provides indications for a larger number of such proteins, for example, through multiple subcellular locations[35] or RNA binding[41], even though such data do not prove the existence of additional, non-canonical functions, and may result in overestimates (see ref. 36 and [42] for subcellular locations). Thus, the non-moonlighting protein families we used are also likely to contain some multifunctional proteins, and in practice, at least for some orthogroups, we most likely compare proteins with few vs. many functions rather than proteins with a single vs. multiple functions.

Moonlighting proteins can perform different functions by several different mechanisms, of which change in the oligomeric status is just one. The very weak association between functional variability and QS variability is nevertheless somewhat surprising, given the large number of experimental studies demonstrating the effect of QS on biochemical function. This suggests that, in most cases where the non-canonical function of moonlighting proteins is also associated with a change in QS between species, it is not so much moonlighting that drives the QS change, but moonlighting proteins rather utilise preexisting QS variability, which, initially, in the early stages of its evolution, is likely to be largely neutral. However, our data also indicates that within species, at least in some cases, moonlighting is likely to drive diversification of QS (Fig. 4H) and result in multiple functional oligomers for the same protein.

Taken together, our findings indicate that previous observations suggesting a neutral-like evolution of homooligomers[3,4,12,17,22] are not invalidated by moonlighting, and provide support for the hypothesis that for many homomers QS evolution is a stochastic process, that might be adaptive in shaping the biophysical characteristics of the cell and its homeostasis[4] (i.e., diffusion rates, cytoplasm fluidity), rather than by affecting the biochemical function of the protein.

## Methods

### Data sources

The proteins of the analyses were selected with a largely similar methodology as in ref. 4. We used UniProt proteins that have a crystal structure in the PDB, and meet the following criteria: (1) their quaternary structure is either a homomer or monomer; (2) their structure has resolution better than 3 Å; (3) the PDB structure contains more than 80% of the UniProt sequence; (4) have a minimum length of 100 amino acid residues. Signal and transit peptides were excluded from the protein sequence. Quaternary structure was determined using the first biounit of the entries. For proteins with multiple entries, the one with the largest number of subunits and coverage was used, and proteins that have also heteromer entries were not included. PDB entries with multiple biounits, where the biounits have different quaternary structure, were not used. We also excluded PDB entries that originate from viruses, have chimeric sequences, or form fibrils. Additionally, sequences annotated as antibodies (using the Structural Antibody Database[43] and GO annotations) and MHC proteins were not included in the analyses. This has resulted in 21,341 proteins (homomers and monomers) where the quaternary structure is known, i.e., have at least one high coverage entry in the PDB.

The list of moonlighting proteins was obtained from two sources: the MoonProt database[33], which list most known moonlighting proteins with experimental support, and the MoonDB[34], which contains manually curated and predicted entries, resulting in a list of 759 proteins (excluding viral proteins), of which 433 are present in the orthogroups (Supplementary Data 2), and 336 have experimental support (Supplementary Data 3). The majority of the moonlighting proteins do not have a 3D homomer or monomer structure that meets the above criteria, and some are part of a PDB heteromer. In the case of MoonDB, predicted entries utilise GO terms in the prediction, and, therefore, predicted entries were used only in the analyses that do not involve GO terms. In analyses that do include GO terms (e.g., Figs. 2I–L and 4E–H) predicted MoonDB entries were not used in the definition of moonlighting orthogroups (MOs).

Gene Ontology and Reactome terms, and EC numbers were obtained from the UniProt annotation of proteins. GO and Reactome terms were post-processed using the GO and Reactome term hierarchies (go.obo and ReactomePathwaysRelation.txt files) using the "is_a" and "part_of" relations (for GO), and terms that are not independent, i.e., are in an ancestor-descendant relationship, and are on the same pathway to the top-level term were filtered out, and only the lowest level term was kept. (For example, from the GO Cellular Component terms "mitochondrion" and "mitochondrial inner membrane" only the latter was kept). GO term pairs that connect separate ontologies (e.g., molecular function with biological process) were not used in the filtering.

Human protein-protein interaction terms were obtained from Bio-GRID v.4.4.241. Only interactions between human proteins were used, human-virus interactions were excluded. Subcellular locations[35] of 13,534 human proteins from the Human Protein Atlas were downloaded from https://www.proteinatlas.org/. All locations were used, including the predictions of secreted proteins, except for Supplementary Fig. 4C and D, where locations labelled as uncertain were excluded.

### Orthogroup identification

Orthogroups were identified with the eggNOG-mapper (v2.1.12) tool[30], online, using the 21341 proteins (see above) with known quaternary structure. The default settings were used, except that the queries were realigned to the Pfam database. Proteins were assigned to an orthogroup using their lowest, root-level annotation of eggNOG-mapper, and we required that proteins with the same root annotation have also shared Pfam domains. (Sequence overlaps between the different Pfam domains were permitted.) This procedure usually results in reasonably high tertiary structure similarity within the orthogroups (TM-score larger than 0.5– 0.6, see ref. 4), thus quaternary structure variability within the orthogroups is unlikely to be dramatically affected by tertiary structure variability. The initial number of ~21k proteins was reduced to 19,783 in the process, due to either not having

eggNOG-mapper hits or Pfam domains, and redundancy filtering (i.e., different UniProt IDs with the same sequence). Altogether, we identified 4181 orthogroups, of which 474 have 10 or more proteins (Fig. 2C, D).

In addition to the proteins with known quaternary structure, we also identified the orthogroups of a dataset containing all the 759 moonlighting proteins and the 21341 proteins above. The orthogroups were identified with the same method as above, and two groups were identified in them, one containing the orthogroups with at least one moonlighting protein, and one with no moonlighting proteins. The shared orthogroups of these two sets were used in the downstream analyses, using only the proteins with known quaternary structure (see Fig. 2A vs. 2B).

### Phylogenetic analyses and ancestral state reconstruction

In every orthogroup with 10 or more proteins, we reconstructed the phylogeny of the proteins and the evolution of their quaternary structure using a two-step procedure. Protein sequences were aligned with Mafft-DASH[44] (v7.520), which also utilises the 3D structures of the proteins. The L-INS-I method was used, with the --maxiterate 1000 --localpair flags. Signal and transit peptides were removed from the sequences before the alignment. The alignments were subsequently processed with IQtree v2[45] to obtain maximum-likelihood phylogenetic trees. Next, to reduce any errors in the tree topologies caused by proteins that evolve significantly faster than the rest of the tree (either due to long-branch attraction or the subsequent rooting), for every protein of the tree its distance from all other proteins was calculated with the "cophenetic.phylo" function of the APE R package (v5.7)[46]. Proteins that are more distant than the average +3 standard deviations were excluded; the filtered set proteins was subsequently realigned, and the trees were rebuilt with the same method as previously. (While this filtering step increased the robustness of the phylogenetic analysis, it has no effect on any of the conclusions.) Next, the trees were rooted with the Minimal Ancestor Deviation method (v2.2)[47]. The most likely quaternary structure of the internal nodes of the tree and its root were estimated with two methods; discrete maximum likelihood (using the "ace" function of the APE R package, with the equal rates model; Fig. 3) and maximum parsimony (using the "phangorn" R package, with the ACCTRAN method; Supplementary Fig. 5). For a fraction of the trees the quaternary structure of the root could not be estimated with a probability higher than 51%, these were excluded from the analyses (Figs. 3 and S5 and S6). The frequency of QS changes in the trees was determined as the number of parent-child nodes where the QS is not the same in the two, divided by the total number of node pairs in the tree, excluding pairs where for at least one of the nodes QS could not be estimated (i.e., its probability was lower than 51%).

### Identification of surface and interface residues, interface overlap, and conservation

Interfaces, their overlaps, and their conservation were identified as described previously in ref. 4. For every protein, we identified the solvent accessible surface area (SASA) in its representative protein complex, and in its monomeric subunits, using FreeSASA v.1.1[48]. Interface size of the complexes was determined as the difference of the SASA of the monomeric subunits and the complex, divided by the number of subunits. Interface residues were defined as residues with relative solvent accessibility (RSA) being 0.2 or above, compared to the full amino acid areas as defined in ref. 49, and with a minimum 10% change in SASA compared to the SASA of the residue in the monomeric subunits. Surface residues were defined as residues that are not part of the interface and have RSA above 0.2.

Interface overlap within orthogroups was calculated with TM-align[50] (v.20190822). We made all possible pairwise comparisons of homomers within an orthogroup, and using the main output of TM-align, we identified the structurally aligned residues in them. Interface overlap was calculated as the ratio of the number of interface residue pairs that are structurally aligned, and the number of residues in the smaller interface. Averages were calculated using all pairwise comparisons in the orthogroup. Structures with the

same number of subunits were considered topologically different if the overlap of their interfaces was lower than 0.5.

Conservation of interfaces was calculated using the summary files downloaded from the ConSurf database[40]. For every homomer structure, the average conservation of its interface and surface residues was calculated, and their difference was used in the analyses. The comparison of interface conservation was performed in two ways, using the average conservation and average interface size of every orthogroup (Fig. 4I–K), and using clustering (Supplementary Fig. 12). In the latter case, clusters with minimum 30% sequence identity were defined with MMseqs2[51] (using the flags: --alignment-mode 3 --min-seq-id 0.3 -c 0.9 --cov-mode 1 --cluster-mode 2 -s 8.5). The cluster centroids were used in the analysis.

### Measures of quaternary structure variability

We used four measures to describe the variability and characteristics of quaternary structure in the orthogroups: (1) Average number of subunits per protein. (2) QS diversity, measured as the number of different QS topologies. Besides differences in subunit number (monomer, dimer, tetramer, etc.), differences in interfaces were also taken into account; homomers with the same number of subunits but different interfaces (with interface overlap <0.5) were treated as separate topologies. (3) Shannon index, calculated as $-\sum_i^n p_i \ln(p_i)$, where $p_i$ is the frequency of a particular QS in the orthogroup, and $n$ is the number of different QS topologies. Unlike the simple diversity measure above, the Shannon index also takes into account how common the different QS topologies are in the orthogroup (e.g., 1 monomer and 9 dimers vs. 5 monomers and 5 dimers, where the latter has a higher Shannon index), thus it provides a better estimate of QS diversity. (4) Number of different QSs per protein. In the previous metrics, we used one representative structure for each protein. However, many proteins have several entries in the PDB, which sometimes can have different quaternary structures. We determined the number of different QSs for every protein, with a similar method as described above (using TM-align), taking into account differences in interfaces, thus entries with similar number of subunits but different interfaces were treated as different QSs. The first biounits of all known entries of the proteins were used, even if they had lower coverage or resolution. It is important to note that this metric is error prone, as for many proteins that are known to function as a multimer, and do have multimeric entries, monomeric entries can also be present in the PDB.

### Statistics and reproducibility

All statistical analyses were performed with R (v.4.1.2) and were plotted with the ggplot2 R package. Boxplots indicate the median, 25–75% interquartile range (IQR), whiskers indicate up to 1.5 * interquartile range from the hinge, notches are defined as 1.58 * IQR/sqrt(n). Datapoints beyond whiskers are plotted as outliers. The Benjamini-Hochberg method (FDR) was used to correct for multiple testing; we used all $p$ values of the study in the correction. Both the uncorrected and corrected $p$ values are reported on the figures, the latter ones in parentheses. All statistical tests are two-sided.

### Reporting summary

Further information on research design is available in the Nature Portfolio Reporting Summary linked to this article.

### Data availability

Data necessary to reproduce all figures and Supplementary Figs. is available at Zenodo (https://doi.org/10.5281/zenodo.17378929). All other data are available from public databases or the corresponding author on reasonable request.

### Code availability

Code necessary to reproduce all figures and Supplementary Figs. is available at Zenodo (https://doi.org/10.5281/zenodo.17378929).

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

## Acknowledgements

We thank Constance J. Jeffery for providing the list of proteins in the MoonProt database. The study was supported by a Biotechnology and Biological Sciences Research Council (BBSRC) grant number BB/Y000730/1, Swedish Research Council (Vetenskapsrådet) grant no. 2023-04254, 2019-05356, Formas grant 2019-01403 (A.Z.), and Marius Jakulis Jason Foundation scholarship (A.Z.). This work used the resources of the Computational Research, Engineering and Technology Environment (CREATE) of King's College London.

## Author contributions

G.A. conceived the project, designed and performed the analyses, and wrote the first version of the manuscript. A.Z. contributed to the experimental design, red and edited the manuscript, and provided funding.

## Competing interests

The authors declare no competing interests.
