## [Transparent Peer Review file · Communications Biology]

Structural variability of multifunctional proteins indicates frequent stochastic evolution of protein oligomers

Corresponding Author: Dr Gyorgy Abrusan

This manuscript has been previously submitted at another journal. This document only contains information relating to versions considered at Communications Biology.

Version 0:

Reviewer comments:

Reviewer #1

(Remarks to the Author)

This is quite an interesting follow paper on an earlier paper that discusses the evolutionary causes of variation in quaternary structures.

In this paper the authors investigate whether seemingly pointless variation in self-assembly state could be due to moonlighting functions, which utilise different stoichiometries. The authors ultimately find no support for this theory, which does not diminish the impact of this work. Rather, it further strengthens the idea that variation in quaternary structure might be largely neutral.

As also reviewed (and enjoyed) the earlier manuscript on this topic, I only have a small number of comments, which will sound familiar to the authors.

1) The authors will remember my objections to root reconstructions using likelihood methods. The (unkownable) position of the root node along the root branch will influence the inference at the root especially in cases when the inference is not decidable by parsimony. I suggest the same remedy as last time: Will the authors please acknowledge this problem and state how many inferences would differ when using parsimony instead.

2) Will the authors please label their phylogenetic trees clearly. It is important that the reader can assess the plausibility of the topologies. Figure three needs taxon labels, and I would much prefer full species names in Figure 1 (and 3).

Reviewer #2

(Remarks to the Author)

In their manuscript, Abrusán and Zelezniak use the phenomenon of moonlighting proteins to explore an important question in the field of protein evolution: is the evolution of new quaternary structures driven by positive selection or by neutral processes? I think this is a very clever way of approaching this question, and I enjoyed reading the manuscript, which I found was well written with informative figures.

Much of the results show no difference between the properties of non-moonlighting and moonlighting proteins in regard to quaternary structure evolution, which the authors use to support the hypothesis that these changes can often be driven by neutral processes. However, another interpretation is that the subjectivity inherent in classifying a protein as "moonlighting" obscures any specific properties that might be different. I think the authors should:

1) Give more information on how MoonDB and MoonProt classify proteins as moonlighting. The criteria should be explicit and discussed in the paper, as they will greatly influence the composition of both protein groups.

2) Discuss what they would consider moonlighting or not: Is a multi-domain enzyme performing multiple steps in a biochemical pathway automatically moonlighting? If not, how different do two protein functions need to be for it to count as moonlighting? In Figure 2G, the authors show that moonlighting orthogroups tend to have a higher number of Pfam annotations: how much of the results are driven by this property? I would expect many properties (ex., number of PPIs) to

correlate strongly with the number of annotated domains.

3) Do all moonlighting functions identified so far significantly contribute to cell fitness? If not, how does that impact the hypothesis that positive selection can lead to new quaternary structures in these proteins?

Again, I think this is a very interesting idea, but the fact that what counts as moonlighting vs what does not is poorly defined (in general, not especially in this manuscript) makes the interpretation of the author's results challenging.

Reviewer #3

(Remarks to the Author)

This manuscript deals with the hypothesis of neutral evolution of quaternary structure in relation to moonlighting. I find neutral evolution a challenging topic because it may not lead to a "this is it" plot. I think the authors do a good job of trying to explore reasonable possibilities given the existing data. This is a valuable contribution to the literature and should be published given the following concerns are reasonably answered in a revised version.

Concerns related to study design and discussion of the results:

1. The authors indicate in the introduction that proteins can "moonlight" via several different mechanisms, including changes in oligomerisation. Their study is a large scale one designed around variables that are known for many proteins, such as GO terms and quaternary structures as reported in the PDB. This potentially misses phenomena that are poorly captured by these variables, in the way they are usually measured. The authors should explore this possibility by using the information from small scale, detailed mechanistic studies on individual proteins. For example, they could go deeper into the literature of proteins that moonlight via changes in oligomerization and discuss whether their large scale approach would capture paradigmatic cases.

2. The authors focus on the evolution of homologous groups. A similar set of questions may be asked for the relationship between moonlighting and quaternary structure in the evolution of paralogous groups. I would like the authors to discuss this in a broad sense. For example: are there previous reports on particular protein systems? Is neutral evolution more or less likely in paralogous versus homologous groups? How confident can we be on the discrimination of orthology versus paralogy in the datasets used in the manuscript? How can this influence the results and their interpretation?

3. Articles 10.1038/s41598-017-05084-8 and 10.1002/prot.25065 are very close to the topic of the manuscript and should probably be included in the discussion

Concerns related to data analysis:

3. The authors perform dozens of statistical tests. Thus, they should correct their p-values for multiple testing.

4. The authors correctly indicate that erroneous quaternary structure in some PDB entries may skew their results. It would help if they perform calculations on synthetic datasets, using estimated uncertainties in quaternary structure, to assess how large this effect may be.

Concerns related to writing:

5. Several panels of Figure 2 are referred to in the text as belonging to Figure 1.

6. anel 2J is erroneously referred to in the text as panel 2I.

7. Reading is at some points confusing due to multiple abbreviations (QS, OG, HPA, MO). I suggest the authors keep the one used more often and use the full words for the rest.

Version 1:

Reviewer comments:

Reviewer #1

(Remarks to the Author)

The authors have satisfactorily revised the manuscript. I do request that they include the parsimony analysis. I will stress yet again that root inferences that are ambiguous for parsimony cannot be magically saved using a likelihood approach. Any additional root confidence in the likelihood reconstruction comes entirely from the completely arbitrary placement of the root node along the root branch. This parameter is not known and the authors should not fool themselves into thinking their likelihood reconstructions are in any way more reliable. It is simply bad mythology. When the authors claim parsimony is less accurate than likelihood, the papers that make this statement deal with problems very different from a root reconstruction, so these arguments simply do not apply. The only way to make a root node inference without making up arbitrary parameter values (i.e. position of the root node along the root branch) is parsimony.

I do not need to see this paper again and trust that the authors will present their parsimony results.

Reviewer #2

(Remarks to the Author)

The authors have addressed all my comments satisfactorily.

Reviewer #3

(Remarks to the Author)

The authors have answered all my concerns in a satisfactory manner with a combination of reasoning, additional experiments and manuscript modifications. I thank them for their work and believe the work is now ready for publication.

Reviewer #1 (Remarks to the Author):

This is quite an interesting follow paper on an earlier paper that discusses the evolutionary causes of variation in quaternary structures.

In this paper the authors investigate whether seemingly pointless variation in self-assembly state could be due to moonlighting functions, which utilise different stoichiometries. The authors ultimately find no support for this theory, which does not diminish the impact of this work. Rather, it further strengthens the idea that variation in quaternary structure might be largely neutral.

As also reviewed (and enjoyed) the earlier manuscript on this topic, I only have a small number of comments, which will sound familiar to the authors.

1) The authors will remember my objections to root reconstructions using likelihood methods. The (unkownable) position of the root node along the root branch will influence the inference at the root especially in cases when the inference is not decidable by parsimony. I suggest the same remedy as last time: Will the authors please acknowledge this problem and state how many inference would differ when using parsimony instead.

We repeated the analyses of Figure 3 using maximum parsimony as suggested by the Reviewer. We found that the observed patterns are the same, and using MP does not influence any of the conclusions (see below). The only difference is that the number of trees where the quaternary structure of the root node cannot be identified is higher, and in consequence, the sample sizes on the figure are somewhat lower. (Please see also the next paragraph, for additional analyses).

2) Will the authors please label their phylogenetic trees clearly. It is important that the reader can assess the plausibility of the topologies. Figure three needs taxon labels, and I would much prefer full species names in Figure 1 (and 3).

We added a taxonomy column to Figures 2A and 2B, that indicates the domain of the protein (i.e. Bacteria, Archaea, Eukaryota). The schematic trees on Figures 3A and 3B are for demonstration purposes only, and they have identical topologies. Their purpose is to show an intuitive case of a homomer root and a monomer root; their topologies are identical, only the quaternary structure assignments of their nodes are different from each other. We believe there is no benefit in adding labels. (These trees are adapted from a branch of COG0005, [purine nucleoside phosphorylase], but the QS of the leaves is arbitrarily set.)

However, trees A and B of Figure 2 did contain an error: the proteins AOA6A5BXC3_NAEFO and Q9NDG0_TRIVA are enolase-homologs from amoebae (*Naegleria fowleri* and *Trichomonas vaginalis*), which were acquired through horizontal transfer from a bacterium. This resulted in an unusually long branch length, which affects the phylogeny both through the positioning the root, but also likely to influence the results through long-branch attraction (see <https://doi.org/10.1016/j.cub.2025.06.045> for an example and description of such effects). Such interdomain horizontal transfers are generally rare; nevertheless, to test whether our conclusions on Figure 3 are influenced by proteins that evolve at a much higher rate than the rest of the tree, we repeated the analysis on Figure 3, excluding proteins with long branches. We made two different analyses, and excluded branches that are (1) longer than the average plus three standard deviations of the distances from other nodes (present in 77 trees); or (2) longer than the average plus two standard deviations of the distance from other nodes (present in 295 trees).

The results show that our conclusions do not depend on the presence/absence such relatively fast evolving proteins (see below). However, their exclusion slightly increases the number of orthogroups where the quaternary structure of the root can be determined, both with the maximum likelihood and the maximum parsimony methods. This makes the analysis more robust (with somewhat higher statistical power), and in the main text, we decided to use the trees where the proteins with the longest branches (above average + 3 SD of the distance from other proteins) were excluded.

The trees from Figure 2 with a taxonomy column added (without AOA6A5BXC3_NAEFO [panel A] and Q9NDG0_TRIVA [panel B], which were more distant from other proteins than the average + 3 SD):

Figure 3, excluding long branches from the trees that are longer than the average + 3 SD of the distance from all other proteins:

Figure 3, excluding long branches of the trees that are longer than the average + 2 SD of the distance from other proteins:

Overall, the more long, outlier branches are removed, the better maximum parsimony (MP) approximates maximum likelihood (ML), which is not surprising, given that MP does not use branch lengths in ancestral state estimates. MP has the tendency to produce a somewhat higher number of trees with a monomer root (2% to 5% higher), nevertheless, the correspondence between the two methods is good, and none of these methodological differences influence the conclusions. We think the comparison of the two methods is relevant for review purposes (i.e. checking whether the conclusions are the same or not, and how large the difference is), however ML is the more accurate, while MP is generally considered to be only an approximate method, thus we decided to use only ML in the manuscript.

Reviewer #2 (Remarks to the Author):

In their manuscript, Abrusán and Zelezniak use the phenomenon of moonlighting proteins to explore an important question in the field of protein evolution: is the evolution of new quaternary structures driven by positive selection or by neutral processes? I think this is a very clever way of approaching this question, and I enjoyed reading the manuscript, which I found was well written with informative figures. Much of the results show no difference between the properties of non-moonlighting and moonlighting proteins in regard to quaternary structure evolution, which the authors use to support the hypothesis that these changes can often be driven by neutral processes. However, another interpretation is that the subjectivity inherent in classifying a protein as “moonlighting” obscures any specific properties that might be different. I think the authors should:

1) Give more information on how MoonDB and MoonProt classify proteins as moonlighting. The criteria should be explicit and discussed in the paper, as they will greatly influence the composition of both protein groups.

We added this to the manuscript. MoonProt and MoonDB are different; MoonProt is a manually curated database maintained by Constance Jeffery, and in theory it lists most proteins for which experimental evidence for moonlighting is available. The majority of moonlighting proteins in our dataset originate from MonProt. MoonDB is a small database focusing on a number of model organisms, and it contains both manually curated and computationally predicted entries. The computational prediction utilises the dissimilarity of Gene Ontology terms for detecting moonlighting proteins (<https://academic.oup.com/nar/article/47/D1/D398/5146199>). Since on several figures we use GO terms to estimate the number of functions, this results in a non-independence problem for some proteins, and in the revision, we decided to exclude the predicted moonlighting proteins from these analyses. (This was an oversight in the initial analysis, the kind of thing one wonders “how on earth could I overlook this”.)

Thus, in all analyses/figure panels that do not utilise GO terms, we use the combined dataset from MoonProt and MoonDB (as in the initial version of the manuscript) in the definition of moonlighting orthogroups, to have the highest possible statistical power, because the main claim of the manuscript is the lack of QS difference between moonlighting and non-moonlighting orthogroups. However, in the analyses/figure panels that do use GO-terms as a variable (and Reactome terms, for example Figure 2I-L, Figure 4E-H), we use MoonProt and only the manually curated proteins from MoonDB to define moonlighting orthogroups, but not the computationally predicted proteins from MoonDB (unless they are also present in MoonProt). Previously we used the full (MoonProt + MoonDB) set in all analyses, we believe the present approach is the correct one.

The change results in a ~20% reduction in the number of the moonlighting orthogroups in these GO-term related analyses, while the number of non-moonlighting orthogroups remain unchanged; orthogroups with only predicted moonlighting proteins from MoonDB were not

included in them. The change does not affect any of the conclusions; significant differences between GO-term numbers in moonlighting and non-moonlighting orthogroups remain significant (e.g. Figure 2I-L and Figure 4E-H), but the effect sizes (and in consequence p-values) are somewhat weaker. However, they are still clear, typically p-values are in the range of 10^{-6} or below.

2) Discuss what they would consider moonlighting or not: Is a multi-domain enzyme performing multiple steps in a biochemical pathway automatically moonlighting? If not, how different do two protein functions need to be for it to count as moonlighting? In Figure 2G, the authors show that moonlighting orthogroups tend to have a higher number of Pfam annotations: how much of the results are driven by this property? I would expect many properties (ex., number of PPIs) to correlate strongly with the number of annotated domains.

In theory, moonlighting proteins have several unrelated functions, for example enzymatic and chaperone. Thus, an enzyme that has one catalytic site which can catalyse multiple different reactions would not be considered moonlighting, while an enzyme with two different, unrelated catalytic sites would be, although in practice, in most cases moonlighting proteins are assumed to have completely unrelated functions, for example enzymatic and chaperone or something else.

The question of how different two protein functions need to be, to be considered moonlighting is in practice very difficult to answer. The literature is not consistent in the question and this is why the estimated numbers of moonlighting proteins vary substantially, from a few percent (2-3%) of the genome to a significant fraction, that can reach 40%. (Some studies automatically assume that proteins with multiple cellular locations are moonlighting.)

In the manuscript, we made no attempt to provide a formal definition of moonlighting. Instead, we simply used presence in these two databases as the criterion for moonlighting, along with homology within an orthogroup (thus, in orthogroups containing at least one moonlighting protein, we assumed that other proteins are also moonlighting). The justification for this is shown in Figure 2, which depicts the phylogeny of Enolases, where only a few moonlighting enolases have a known structure (Figure 2A). However, the total number of known moonlighting enolases is much higher, and importantly, they are essentially randomly distributed across the tree (Figure 2B).

The annotation of PPIs varies across different species; humans are probably one of the best-annotated ones. As suggested by the reviewer, we checked whether there is a correlation between the number of different Pfam domains in a protein and the number of PPIs. There is, nevertheless it is not a very strong one (see below), and it does not explain the difference between moonlighting (MOs) and non-moonlighting orthogroups (non-MOs), MOs have consistently more PPIs than non-MOs, particularly in proteins with a single Pfam domain:

As MoonDB utilises also PPIs in the prediction of moonlighting proteins (although it relies primarily on GO-term dissimilarity), we also checked this using only proteins from the MoonProt database and curated MoonDB proteins (i.e. moonlighting orthogroups were redefined using this reduced set). The pattern is not different:

Taken together, the number of different Pfam domains does not seem to be the main driving force of the number of PPIs.

3) Do all moonlighting functions identified so far significantly contribute to cell fitness? If not, how does that impact the hypothesis that positive selection can lead to new quaternary structures in these proteins?

Our paper relies on the assumption that moonlighting functions of proteins contribute to fitness. However, we think the very definition of function implies a contribution to fitness, and traits without fitness a contribution should not be considered functions.

It is very unlikely that the fitness effects were directly measured for all (or most) moonlighting proteins with experimental support. However, one very interesting study (Espinosa-Cantu et al. 2018, <https://academic.oup.com/genetics/article/208/1/419/6066478>) shows that in the case of certain yeast enzyme knockouts, in approximately one third of the cases growth could be restored by catalytically inactive versions of the enzyme, indicating the non-catalytic functions do have fitness effect, and that it was primarily not the loss of enzymatic activity that resulted in reduced growth. However, most yeast gene knockouts do not have well-measurable phenotypic consequences in the lab, so measuring actual contribution to fitness can be challenging even in yeast.

Overall, I think in the case of conservative estimates, which rely primarily on experimental characterisation of moonlighting functions, a contribution to fitness is very likely in most cases.

In the case of relaxed estimates, which sometimes assume that the occasional RNA binding or variable subcellular location automatically implies functional variability, such a contribution to fitness is frequently questionable. Our analysis uses a conservative set of moonlighting proteins.

Again, I think this is a very interesting idea, but the fact that what counts as moonlighting vs. what does not is poorly defined (in general, not especially in this manuscript) makes the interpretation of the author's results challenging.

We agree that this is a problem of the field, but mainly in the “upper end” of the datasets, i.e. studies that claim that a relatively high fraction of the proteome is moonlighting, based on indirect data like multiple subcellular locations (which is also likely to be substantially affected by randomness). In the case of experimentally characterised proteins that we use, this is unlikely to be a major source of error.

Reviewer #3 (Remarks to the Author):

This manuscript deals with the hypothesis of neutral evolution of quaternary structure in relation to moonlighting. I find neutral evolution a challenging topic because it may not lead to a "this is it" plot. I think the authors do a good job of trying to explore reasonable possibilities given the existing data. This is a valuable contribution to the literature and should be published given the following concerns are reasonably answered in a revised version.

Concerns related to study design and discussion of the results:

1. The authors indicate in the introduction that proteins can “moonlight” via several different mechanisms, including changes in oligomerisation. Their study is a large scale one designed around variables that are known for many proteins, such as GO terms and quaternary structures as reported in the PDB. This potentially misses phenomena that are poorly captured by these variables, in the way they are usually measured. The authors should explore this possibility by using the information from small scale, detailed mechanistic studies on individual proteins. For example, they could go deeper into the literature of proteins that moonlight via changes in oligomerization and discuss whether their large-scale approach would capture paradigmatic cases.

This is a bit trickier than it seems at first, because our main conclusion is that there is no clear difference in quaternary structure evolution between moonlighting proteins. Nevertheless, we acknowledge the point of the Reviewer and have performed the following checks, as suggested (see below).

There are two main ways oligomerisation can contribute to moonlighting: quaternary structure can be variable within species, or between species. For the first case (proteins that have multiple different structures in the same species) paradigmatic cases are peroxidase, which is a dimer as an enzyme and a decamer as a chaperone, porphobilinogen synthase, a morpheein that can be a dimer, hexamer or octamer, and glyceraldehyde-3-phosphate dehydrogenase, which can be a monomer or tetramer depending on its catalytic function. (see Jeffery 1999, <https://pubmed.ncbi.nlm.nih.gov/10087914/>; Liu and Jeffery 2020, <https://pubmed.ncbi.nlm.nih.gov/32751110/>). We highlighted these proteins on Figure 4 (see below, black symbols), which shows that their orthogroups are characterised by higher average nr of subunits (panel A, note that we use here the largest structures for every protein), and a somewhat higher numbers of different topologies per protein (panel D), but not by higher QS diversity within orthogroups (panels B and C). Additionally, the number of their molecular function GO terms is the same as in other moonlighting orthogroups.

For QS variability between species, perhaps the “best” paradigmatic cases are proteins of glycolysis. These proteins were used by the landmark Lynch 2013 paper (<https://www.pnas.org/doi/10.1073/pnas.1310980110>) to illustrate the case of QS variability of enzymes between different taxa. These enzymes are generally thought to have stable structures within a species (the exception is Glyceraldehyde phosphate dehydrogenase, which was included above); nevertheless, all of them are known to moonlight, and in species with different moonlighting functions, they can have different quaternary structures. (See reviews for aldolases, <https://pubmed.ncbi.nlm.nih.gov/34458323/>; and hexokinases <https://pubmed.ncbi.nlm.nih.gov/34235183/>.) So, for example, human aldolase is a tetramer (and besides being an enzyme, it is involved in several, such as, cancer-related processes), while in other, say, prokaryotic species with different moonlighting functions it can have different topologies, e.g. dimers or decamers.

However, highlighting these nine enzymes in Figure 4 (see below, black symbols) indicates that, despite being very intensively studied, none of their quaternary structure parameters show a convincing difference from non-moonlighting orthogroups (especially if orthogroup size is taken into account), even though there is a very clear difference in the number of GO Molecular Function terms (panels E-H).

So, we think the Lynch 2013 PNAS paper is not invalidated by moonlighting, the QS variability observed in these proteins does not indicate that it is shaped by different factors than in the case of non-moonlighting orthogroups, or, as we suggest in the paper, moonlighting does not drive, but rather utilise preexisting QS variability, which, at least initially is the result of stochastic events. (This could also explain why moonlighting functions within the same orthogroup are frequently not conserved across species.)

2. The authors focus on the evolution of homologous groups. A similar set of questions may be asked for the relationship between moonlighting and quaternary structure in the evolution of paralogous groups. I would like the authors to discuss this in a broad sense. For example: are there previous reports on particular protein systems? Is neutral evolution more or less likely in paralogous versus homologous groups? How confident can we be on the discrimination of orthology versus paralogy in the datasets used in the manuscript? How can this influence the results and their interpretation?

In general paralogs are expected to experience relaxed purifying selection compared to orthologs, and should evolve/change faster. We do not discriminate between paralogs and orthologs within an orthogroup, we use all available proteins from the PDB in the analysis that pass structural criteria like high coverage of UniProt sequence, or reasonable ($<3 \text{ \AA}$) resolution. So, if a human gene has a homolog for example in E. coli we use it, even if it is not a direct one-to-one ortholog but a paralog (i.e. the gene underwent duplication in E. coli, and the paralogous copy has a structure in the PDB). In theory this can affect QS diversity, as orthogroups with more conserved proteins are likely to have less diversity, and highly conserved proteins are generally enriched among moonlighting proteins (see Singh and Bhalla 2020, <https://pubmed.ncbi.nlm.nih.gov/32870732/>). To test for this, using the 474 orthogroups which do have a phylogenetic tree, we examined whether there is any relationship between the average phylogenetic distance of the proteins in the orthogroup and the diversity of the quaternary structure. (See the figure below. We do not use raw sequence similarity, because in many orthogroups the proteins are distant homologs, and it is generally low, frequently below 30%, which leaves very little space for measuring differences because random protein sequences have typically 15-20% sequence similarity).

The results show that there is no difference in the diversity between MOs and non-MOs, except for panel A, which, similarly to panels E and F of Figure 2, indicates that in orthogroups of 10 or more proteins homomers are more frequent in MOs than non-MOs. Using two covariates, the

average distance in the tree plus orthogroup size results in the same pattern, lack of difference between MOs and non-MOs. We also found no difference in the average interface conservation of MOs and non-MOs.

We see this as an important control, and we added this figure (with some modifications) to supplementary figures as Figure S9.

3. Articles 10.1038/s41598-017-05084-8 and 10.1002/prot.25065 are very close to the topic of the manuscript and should probably be included in the discussion.

We added these papers to the manuscript as suggested.

Concerns related to data analysis:

3. The authors perform dozens of statistical tests. Thus, they should correct their p-values for multiple testing.

We added P-values corrected with the Benjamini-Hochberg method (FDR) to each figure in parentheses. However, as the main claim of the manuscript is the lack of a clear difference between MOs and non-MOs, therefore we also provide the uncorrected p-values, which in this particular case are more sensitive and relevant.

Altogether we make 124 statistical tests in the paper, and we could get criticised that by making a large number of tests we artificially reduce our ability to detect significant differences. The uncorrected p-values are unaffected by this, thus we provide both the uncorrected and corrected p-value on every figure.

4. The authors correctly indicate that erroneous quaternary structure in some PDB entries may skew their results. It would help if they perform calculations on synthetic datasets, using estimated uncertainties in quaternary structure, to assess how large this effect may be.

It is not entirely clear to us which datasets Reviewer thinks about; nevertheless, we did perform analyses to check whether QS errors influence the conclusions. Separating crystallographic interfaces from “real”, biological ones is a problem that has been around for decades (e.g. Janin 1997, Ponstingl et al. 2003), and studies that try to separate the two show the same pattern: the likelihood of being crystallographic increases sharply as interface size gets smaller. Baskaran et al, 2014 (<http://www.biomedcentral.com/1472-6807/14/22>) performed a large-scale, PDB-wide analysis of crystallographic interfaces, and their results show that the vast majority of crystallographic interfaces are smaller than 1000 Å² (In 2014 the PDB had already ~100K structures).

This is very much in agreement with my own analyses (Abrusan and Foguet 2023, Figure 2 <https://academic.oup.com/mbe/article/40/4/msad070/7083724>), which show that in the case of homodimers, the rate of coevolution between interface residues drops at about 1000 Å², suggesting that below this threshold, dimers with erroneous, crystallographic interfaces are highly enriched. Also, checking a large number of structures over the years manually is very much in agreement with this, most problematic interfaces are small, below 1000 Å². (Of course, there are exceptions, functional homomer interfaces with area < 1000 Å² do exist. So, to check how QS errors might affect the results, we decided to apply a simple strategy: to assume that all homomers with interface area < 1000 Å² are in fact monomers, and reanalyse the data (Figures S5, S6 and S10). The results obtained with this modified dataset do not change any of the conclusions, so we think our results are robust to QS errors. It is beyond doubt that it is possible to perform a test like this differently, using different tools or datasets, but we find it very unlikely that it would change any of the conclusions.

Concerns related to writing:

5. Several panels of Figure 2 are referred to in the text as belonging to Figure 1.

We corrected this.

6. Panel 2J is erroneously referred to in the text as panel 2I.

We corrected this.

7. Reading is at some points confusing due to multiple abbreviations (QS, OG, HPA, MO). I suggest the authors keep the one used more often and use the full words for the rest.

We agree that too many abbreviations in a paper can be difficult to read. We replaced all “OG” abbreviations with “orthogroup”. Since there are only few HPA (Human Protein Atlas) abbreviations, all in one paragraph, and we think they improve readability, we decided to add the definition of HPA at line 187 shortly before the abbreviations to reduce the cognitive load for the reader (so we do not have to spell out “Human Protein Atlas” several times in the same

paragraph). We keep the QS and MO abbreviations as there are a very large number of them and we think they improve the clarity of the text.